# Mechanical Egg Activation and Rearing of First Instar Larvae of *Sirex noctilio* (Hymenoptera: Siricidae)

**DOI:** 10.3390/insects14120931

**Published:** 2023-12-07

**Authors:** Elmarie van der Merwe, Bernard Slippers, Gudrun Dittrich-Schröder

**Affiliations:** 1Department of Zoology and Entomology, Forestry and Agricultural Biotechnology Institute (FABI), University of Pretoria, Pretoria 0002, South Africa; elmarie.vandermerwe@fabi.up.ac.za; 2Department of Biochemistry, Genetics and Microbiology, Forestry and Agricultural Biotechnology Institute (FABI), University of Pretoria, Pretoria 0002, South Africa; bernard.slippers@fabi.up.ac.za

**Keywords:** egg activation, mechanical pressure, *Sirex noctilio*, Hymenoptera, *Amylostereum areolatum*, artificial diet, rearing, larval development, pine, genetic pest management

## Abstract

**Simple Summary:**

Insects in the order Hymenoptera can produce males from unfertilised eggs. However, unfertilised eggs require a specific signal to become ‘activated’ and start development. Manually activating eggs is advantageous for numerous fields of research, especially for insect pest management, but has not been conducted for many hymenopteran species. The invasive woodwasp, *Sirex noctilio* Fabricius (Hymenoptera: Siricidae), is a hymenopteran pest of pine plantations. In this study, we developed a novel egg activation protocol for *S. noctilio*, allowing eggs dissected from a female wasp to develop further into larvae after activation. These first instar larvae were subsequently reared on an artificial diet. Optimal conditions for the development of activated eggs were determined, which involved treatments with the fungal symbiont of *S. noctilio*, *Amylostereum areolatum* (Russulales: Amylostereaceae). The presence of the fungus did not benefit egg development in an artificial laboratory environment. This study forms the basis for new research on *S. noctilio* that may not only refine our understanding of aspects involving its cryptic lifestyle but also aid in the development of next-generation pest management strategies. In addition, closely related insect species may benefit from the developed protocols.

**Abstract:**

Egg activation is a cellular transition of an arrested mature oocyte into a developing embryo through a coordinated series of events. Previous studies in Hymenoptera have indicated that mechanical pressure can induce egg activation. In this study, we developed the first egg activation protocol for the haplodiploid insect pest, *Sirex noctilio* (Hymenoptera: Siricidae), from two climatically different regions in South Africa to demonstrate the broad applicability of the method. In addition, activated eggs were exposed to three treatments involving water, pine sawdust, and the fungal symbiont of *S. noctilio*, *Amylostereum areolatum* (Russulales: Amylostereaceae), to determine if the symbiotic fungus is a requirement for egg development in an artificial laboratory environment, as the symbiotic fungus has been hypothesised to be necessary for egg and early larval development in a natural environment. A rearing protocol was developed for the first instar larvae using a modified *Anoplophora glabripennis* (Coleoptera: Cerambycidae) artificial diet. A significant difference between the mean survival rates of activated eggs from the two different regions was observed. *Amylostereum areolatum* was shown to be unnecessary for egg survival and adversely affected egg eclosion in an artificial laboratory environment. The maximum larval survival duration on the artificial diet was 92 days. The egg activation and rearing protocol developed in this study enables opportunities for research on the physiology, ecology, symbioses, and genetics of *S. noctilio*, which can be exploited for new genetic pest management strategies.

## 1. Introduction

Egg activation is a process whereby the mature oocyte resumes development towards embryogenesis. Preceding egg activation, the development of a mature oocyte is halted at a species-specific stage of meiosis prior to fertilisation [1,2]. For egg activation to occur, an external trigger is required, which in many sexually reproducing animals is the entry of sperm into the egg [3,4].

Some animals can bypass fertilisation and produce offspring without sperm through a reproductive strategy known as parthenogenesis [5]. Parthenogenesis is a common mode of reproduction in insects and is particularly prevalent in the Hymenoptera (ants, bees, and wasps) [6,7]. This order contains diverse types of parthenogenesis, such as arrhenotokous parthenogenesis (arrhenotoky) and thelytokous parthenogenesis (thelytoky) [8,9]. Arrhenotoky, or haplodiploidy, is predominantly found in Hymenoptera and is characterised by the development of haploid males from unfertilised eggs and diploid females from fertilised eggs [8,9].

Eggs from parthenogenetically reproducing insects require additional external stimuli to be activated [10]. In insect species reproducing parthenogenetically, the meiotic block in eggs can be removed through artificial means, thereby enabling egg activation. Artificial egg activation has been successfully demonstrated in insect species such as *Catrausius morosus* (Phasmatodea: Lonchodidae), *Athalia rosae* (Hymenoptera: Tenthredinidae), *Anopheles stephensi* (Diptera: Culicidae), and *Drosophila* species (Diptera: Drosophilidae). For these species, egg activation was induced using a range of methods, from exposing the eggs to oxygen (*C. morosus*) to placing the eggs in distilled water or a hypotonic (*A. rosae*, *A. stephensi*) or hypertonic solution (*Drosophila* species) [2,11,12,13,14].

For hymenopteran species, especially those where the width of the egg is much smaller than the diameter of the oviposition canal, mechanical stress is likely to initiate egg activation [15]. Egg activation has been achieved in five hymenopteran species where different methods were used to apply mechanical pressure and simulate the oviposition event. In *Pimpla turionellae* (Hymenoptera: Ichneumonidae), eggs obtained via dissection from ovaries were activated using mechanical pressure by forcing the eggs through a narrow, artificial capillary [16]. Similarly, eggs from *Nasonia vitripennis* (Hymenoptera: Pteromalidae) [17] and *Apis mellifera* (Hymenoptera: Apidae) [18] were activated. A slightly different method was used in *Campoletis sonorensis* (Hymenoptera: Ichneumonidae), where the eggs were placed in tubing and gently rolled with a glass rod to induce mechanical stress [19].

Access to developing insect eggs, a good understanding of an insect pest’s biology, and the ability to rear the insect through all life stages are important, particularly for the effective implementation of pest control strategies [20]. For instance, a prerequisite for the implementation of genetic pest control strategies, such as *CRISPR/Cas9* gene drives or precision-guided sterile insect technique (pgSIT), is access to large numbers of eggs. Due to the current nature of insect transformation involving microinjection [21,22], insect eggs are a necessary first step to produce transformed individuals for subsequent releases or establishment of transformed insect populations [22,23]. Successful transformation is largely dependent on the timing of microinjection during egg development, which in insects is the pre-blastoderm phase [22]. In addition, the ability to rear insect pests on an artificial diet for mass production is also necessary and practical to substitute the collection and maintenance of bulky plant resources for insect nutrition [24].

*Sirex noctilio* (Hymenoptera: Siricidae) is an invasive insect pest and is important to control since it causes severe economic losses in exotic pine plantations globally [25]. Apart from larvae damaging the production quality of wood [26], *S. noctilio* is the only known siricid wasp able to kill relatively healthy trees [27]. Adult females oviposit eggs into the host tree along with arthrospores of their fungal symbiont, *Amylostereum areolatum* (Chaillet ex Fr.) Boidin (Russulales: Amylostereaceae), and a phytotoxic mucus [28]. The fungal–mucus complex causes host tree death by weakening the tree’s defence system [29,30]. Larvae lack enzymes to directly digest xylem and depend on *A. areolatum* as an ‘external gut’ to degrade lignocellulosic compounds [31], and the mucus aids in the establishment of *A. areolatum* [32]. The role of *A. areolatum* in *S. noctilio* egg development has received limited attention and has been hypothesised to be required for egg eclosion [33,34].

To date, control methods for *S. noctilio* have relied on the use of silvicultural practices and biological control agents, *Deladenus siricidicola* (Tylenchida: Neotylenchidae) and *Ibalia leucospoides* (Hymenoptera: Ibaliidae) [35], but no work has been conducted exploring the application of genetic pest control methods. This is in part due to the paucity of knowledge available on the early life stages of *S. noctilio*, due to the wasp utilising the host for a large part of its development [36], the inability to obtain large numbers of developing eggs and the absence of an artificial diet for the rearing of the early life stages of *S. noctilio*. Access to *S. noctilio* eggs is challenging as they can be oviposited up to 12 mm into the sapwood of the tree [37]. Resin droplets are often produced at the oviposition site [26,38]; however, the exact site remains difficult to locate due to its small size. Alternatively, eggs can be removed from the ovaries of adult females, but they do not develop and will only produce male offspring as the wasp has a haplodiploid sex-determination system, where males are produced from unfertilised eggs [39].

In this study, we aimed to develop a simple egg activation protocol for *S. noctilio* using microscope slides and eggs from two climatically different regions in South Africa. It is hypothesised that mechanical pressure may be the trigger for egg activation in *S. noctilio* since eggs move through a narrow ovipositor during oviposition. In addition, optimal conditions for egg development and overall survival were tested, including the potential role of the fungus *A. areolatum*. We hypothesised that *A. areolatum* would be beneficial for the development of *S. noctilio* eggs. A second objective was the development of an artificial diet for *S. noctilio* larvae emerging from the activated eggs. This knowledge is important to answer questions on the biology of the organism and for the development and implementation of genetic pest control methods.

## 2. Materials and Methods

### 2.1. Insect Collection

*Pinus* sp. logs (±80 cm in length and a diameter ranging from 10 to 35 cm) infested with *S. noctilio* were harvested from plantations in two climatically different regions in South Africa, namely the Western Cape (winter rainfall region; *P. radiata*) and Mpumalanga (summer rainfall region; *P. patula*) (Appendix A). Different regions were used to evaluate if the egg activation method would be applicable to different *S. noctilio* populations. Logs from each region were stored in an outdoor double-netted cage (100% polyolefin with 29% porosity, Xsect Xtra, Svensson, Sweden and 80% black net, Greenhouse Products [Pty] Ltd., Dynatrade, South Africa) at the Forestry and Agricultural Biotechnology Institute (FABI) Biocontrol Facility, University of Pretoria, South Africa (25°45′09.4″ S 28°15′22.3″ E) at different time points between March 2021 and March 2023. To allow for maximum emergence of individuals, logs were crossed-stacked (Figure 1, step 1) and occasionally rotated. All log ends were painted with a non-toxic waterproof sealer (Watershield clear, Atlas Paints, Pretoria, South Africa) and sprayed with water on a regular basis to maintain adequate internal moisture levels for optimal larval and pupal development. For each trial, 10 *S. noctilio* adult females were collected: Trial-1 (“Egg activation”) females were collected from Batch 1 (Appendix A) from March to April 2021; Trial-2^MP^ (“Egg activation and subsequent development with treatments” for Mpumalanga) females were collected from Batch 2 (Appendix A) between October and November 2022; and Trial-2^WC^ (“Egg activation and subsequent development with treatments” for Western Cape) females were collected from Batch 3 (Appendix A) between February and March 2023.

### 2.2. Dissection and Extraction of Eggs

Female wasps that emerged from pine logs in the outdoor double-netted cage were dissected using Vannas scissors (8 cm, straight, 3 mm blades; World Precision Instruments, Sarasota Center Blvd, Sarasota, FL, USA) (Figure 1, step 1–3). For Trial-1, living wasps and wasps that died within 24 h were dissected, whereas only living wasps were dissected for Trial-2. The eggs from each individual were removed from the ovaries using Super-Fine-5-Swiss pattern forceps (BioQuip, Rancho Dominguez, CA, USA) and placed onto a clear, sterile microscope slide (26 × 76 mm, 1 mm–1.2 mm thick; Labocare, London) in a few drops of SABAX pour water (Adcock Ingram, Midrand, South Africa). Eggs were separated slightly to allow examination under a compound microscope (Nikon Eclipse E200, Nikon Corporation, Tokyo, Japan) to determine whether or not the eggs were parasitised by *D. siricidicola* (Figure 1, step 4). Parasitised eggs were discarded. Non-parasitised eggs were identified by the absence of nematodes and an undigested cytoplasm. Non-parasitised eggs were used for further experiments.

### 2.3. Egg Activation

To determine if pressure is required to activate *S. noctilio* eggs, eggs from 10 unparasitised females from the Western Cape were used (Trial-1). This experiment was based on preliminary work, where a number of different methods were tested to determine which method had the potential to successfully activate *S. noctilio* eggs. Preliminary work using two microscope slides indicated some success in activating *S. noctilio* eggs using a certain level of pressure.

All unparasitised eggs from each *S. noctilio* female were divided equally onto three microscope slides containing SABAX pour water and subjected to either one of the following treatments: (1) no mechanical pressure (control), (2) light mechanical pressure, and (3) substantial mechanical pressure. Light pressure was exerted by placing a microscope slide on top of the first slide containing the eggs, for one minute, creating a ‘double-slide compress’ (Figure 1, step 7). Substantial mechanical pressure was applied by exerting a maximum vertical downward force (±90 N) on the double-slide compress for one minute (Figure 1, step 8). The vertical downward force was quantified by following Newton’s second law of motion (force = mass × acceleration). Pressure was applied on a scale in the same manner as on the double-slide compress, and the average displayed weight was multiplied by Earth’s gravitational acceleration (9.81 m.s^−2^); thus, 9.2 kg × 9.81 m.s^−2^ = 90.25 N.

The double-slide compress was disassembled, and activated eggs remained on the slide with a small amount of SABAX pour water to prevent desiccation. Microscope slides with eggs from all three treatments were placed in Petri dishes (90 mm, JP Last Plastic Products, Roodepoort, South Africa) on Whatman filter paper discs (Grade 201, 90 mm diameter, GE Healthcare Life Sciences, Buckinghamshire, UK) dampened with 2 mL of SABAX pour water and sealed with PARAFILM ‘‘M’’ Laboratory film (Pechiney Plastic Packaging, Inc., Chicago, IL, USA) (Figure 1, step 9). The amount of water on the slide surrounding the eggs was monitored to ensure that the eggs remained hydrated but were not submerged in water. Eggs were kept at room temperature for 11–12 days until larvae emerged. The survival rate was recorded as the percentage of larvae that emerged from a batch of activated eggs.

### 2.4. Egg Activation and Subsequent Development with Treatments

Egg activation treatments were conducted for 10 female wasps from each region, i.e., Mpumalanga (Trial-2^MP^) and Western Cape (Trial-2^WC^) (Appendix A). Unparasitised eggs from individual wasps were placed in a separate Petri dish (90 mm) onto a Whatman filter paper disc (Grade 201, 90 mm diameter) dampened with 2 mL of SABAX pour water. Per female, 150 eggs (with the exception of one female, which only had 120 eggs) were cleaned on the filter paper disc using a fine paintbrush (Princeton Select Round petite, 12/0) to remove any organic material attached to the eggs. The cleaned eggs were placed between two microscope slides in 30–50 µL of SABAX pour water, creating a double-slide compress. Substantial mechanical pressure was applied to the double-slide compress by placing a thumb on each end and pressing down with a maximum vertical downward force for one minute. The double-slide compress was disassembled, and eggs were kept on one of the slides in a small amount of SABAX pour water. Subsequently, the activated eggs were divided equally into three groups (50 eggs per group, with the exception of one female, which had 40 eggs per group). Each group was placed onto a separate microscope slide inside a Petri dish (90 mm) onto a Whatman filter paper disc (Grade 201, 90 mm diameter) dampened with 2 mL of SABAX pour water. Each group received one of the following treatments: (A) 50 µL *Amylostereum areolatum* (CMW 40871) mycelial suspension in SABAX pour water and 1 g autoclaved raw pine sawdust; (B) 50 µL SABAX pour water and 1 g autoclaved pine sawdust; or (C) 50 µL SABAX pour water. The *A. areolatum* mycelial suspension was made by scraping all mycelia off an overgrown *A. areolatum* culture grown on malt extract agar (MEA) (20 g Malt and 30 g purified Agar) for about 30 days using a size 24 surgical blade (Huaian Guangda Medical Instruments Co., Ltd., Huaian, China) and suspending it in 2 mL SABAX pour water inside a Falcon tube (15 mL). The pine sawdust was made by abrading the sapwood of a *P. patula* log from the outdoor netted cage using a Metabo BAE 75 belt sander (75 × 533 mm, 1010 W). The pine sawdust was autoclaved three times inside a 1 L Schott Duran bottle (Duran^®^ Products, Mainz, Germany).

To prevent cross-contamination between treatments, Treatment C was prepared first, followed by Treatment B, and lastly, Treatment A, using sterilised equipment between each treatment. Eggs were fully submerged in each treatment mixture for the entire duration of the trial. Petri dishes were closed using Parafilm and placed in a cake saver (9.5 L ADDISWARE, Western Cape, South Africa) covered in tinfoil inside a dark incubator at 25 ± 1 °C and 52 ± 2% r.h. Moisture levels of the filter paper and moisture around the eggs were monitored regularly and replenished with SABAX pour water when necessary.

### 2.5. Rearing of Sirex Noctilio Larvae, Emerging from Activated Eggs, on a Modified Artificial Diet

Insects mostly require the same type of nutrients (carbohydrates, proteins, lipids, minerals, vitamins, and trace elements); however, the amount can differ between developmental stages of a species and also among species [40]. *Anoplophora glabripennis* (Coleoptera: Cerambycidae), Asian long-horned beetle, and *S. noctilio* are both xylophagous insects [41,42]. Many studies have dedicated research towards developing an artificial diet for *A. glabripennis* [43,44,45,46,47] and, thus, serve as a good starting point to develop an artificial diet for *S. noctilio*.

Natural Resources Canada produces a commercially available diet for *A. glabripennis*, which was originally developed by Keena (2005) [45]. The diet can be ordered online from (http://insect.glfc.cfs.nrcan.gc.ca/cart-panier/diets-dietes.cfm?lang=eng, accessed on 13 August 2023), either already prepared or as separate dry ingredients, with instructions for preparation. Dry ingredients can also be ordered from Sigma-Aldrich (Merck KGaA, Darmstadt, Germany) (Appendix A). In this study, the dry ingredients were ordered to enable customising the recipe for *S. noctilio* and to save on rearing costs by producing the diet locally and on a smaller scale. The modifications made to the diet are described below and summarised in Table 1.

The diet was prepared under sterile laboratory conditions. The Natural Resources Canada recipe ingredient quantities were divided by 32, and the five original ingredient groups were kept the same. Streptomycin sulfate salt and SABAX pour water were added as Group 6 ingredients, and autoclaved pine sawdust was added as Group 7 ingredients. Pine sawdust was obtained by abrading the sapwood of a *P. patula* log from the outdoor netted cage using a Metabo BAE 75 belt sander (75 × 533 mm, 1010 W). Agar and distilled water were autoclaved and allowed to cool to ±40 °C. Group 1, 2, and 4 ingredients were separately mixed and kept in separate glass beakers. Group 5 and 7 ingredients were mixed together. The ingredients from Group 3 were mixed thoroughly in a Petri dish (30 mm). Group 6 ingredients were mixed and added to the cooled agar using a 3 mL syringe and a sterile Acrodisc ^®^ PF syringe filter (0.8/0.2 µm Supor^®^ membrane low protein binding non-pyrogenic, Pall Life Sciences). The agar mixture was poured into a stainless-steel mixing bowl, whereafter, Group 1 ingredients were added and mixed in using an electric hand mixer (Kenwood 280 W) for 10 s. Group 2 ingredients were added to the mixing bowl and combined with an electric hand mixer for 10 s. Group 3 ingredients were added and mixed in with a spoon, followed by Group 4 ingredients. The Group 5 and 7 mixture was added and mixed in with a spoon. Lastly, all ingredients were mixed together with a hand mixer for 10 s. The diet was quickly scooped into Petri dishes (60 mm) and left to cool in a laminar flow hood for about 30 min with lids slightly open until condensation subsided. Petri dishes were wrapped in Parafilm and stored at −20 °C until usage.

Larvae were reared on the above-mentioned artificial diet, which was placed on microscope slides inside Petri dishes (Appendix A). A size 24 surgical blade was used to transfer a 1 × 1 cm piece of diet onto a microscope slide. The surgical blade was then used to make 1 × 0.1 cm groves in the diet surface to mimic galleries made in the host tree. Larvae were transferred into the groves using a soft, fine paintbrush. The microscope slide containing the diet and larvae was placed inside a Petri dish (90 mm) onto a Whatman filter paper disc (Grade 201, 90 mm diameter), dampened with 2 mL of SABAX pour water, and sealed with Parafilm. Petri dishes were placed in a cake saver, covered in tin foil, and placed inside a dark incubator at 25 ± 1 °C and 52 ± 2% r.h. The filter paper inside the Petri dishes was kept damp, and larvae were transferred onto a fresh piece of diet every 14 days.

### 2.6. Statistical Analysis

No statistical analyses were conducted for Trial-1, as two of the treatments had no survival. For Trial-2^MP^ (470 eggs) and Trial-2^WC^ (500 eggs), a Shapiro–Wilk test was performed to determine whether the data were normally distributed. A one-way Analysis of Variance (ANOVA) was used to compare the interaction between the independent variable (treatment) and the dependent variable (survival rate) between each treatment for Trial-2^MP^ and Trial-2^WC^. The data for both Trial-2^MP^ and Trial-2^WC^ conformed to the assumption of constant variance and normality of the ANOVA test. Tukey’s Honestly Significant Difference (HSD) post hoc test was performed to compare pairwise treatment combinations for Trial-2^WC^. A Kruskal–Wallis test is a non-parametric test that can be used to determine if a significant difference exists between two or more independent groups. Therefore, it was used to determine if the overall survival rates between Trial-2^MP^ and Trial-2^WC^ were significantly different since the data did not conform to the assumption of constant variance and normality of the ANOVA test. Differences were considered significant when the *p* value < 0.05. All statistical analyses were conducted using JMP^®^ Statistical Software version 17 (SAS Institute, Cary, NC, USA).

## 3. Results

### 3.1. Egg Activation

No development was observed for any eggs subjected to Treatment 1 (no mechanical pressure—control) and Treatment 2 (light mechanical pressure). However, for Treatment 3 (substantial mechanical pressure), the survival rate ranged from 23.08 to 98.27% with an average of 75.72 ± 9.135% (mean ± SE). Successfully activated eggs could be identified within 24 h after activation by the contraction of the germ band, creating an open space at the anterior and posterior poles of the egg (Figure 2). During subsequent days, the appearance of the yolk changed by gradually moving to the center and membranes forming around it. By day eight, faint segmentation became visible. Brown chitinous mandibles and a pointed terminal spine became visible by day 10, and embryos exhibited movement. Larvae mostly hatched from the posterior end of the egg by tearing the chorion with their spines. Most of the larvae emerged on day 12, but some emerged either a day earlier or one to two days later (Figure 2).

Larvae that emerged from the egg capsule were translucent and segmented with a head distinguishable from other body segments and slightly hooded at the prothoracic segment. Translucent lateral ocelli and brown mandibles were observed, as well as a dark brown pointed spine with setae located above the spine on the ventral side of the last segment. A longitudinal tracheal system was clearly visible, connecting faint spiracles along the lateral side of the abdominal segments. Overall, the shape of the first instar larvae was the same as the more mature larvae.

### 3.2. Egg Activation and Subsequent Development with Treatments

The Shapiro–Wilk test indicated that the data for Trial-2^MP^ (Figure 3) was normally distributed (W = 0.930; *p* < 0.05). Thus, a one-way ANOVA was performed, which indicated that the survival rate across all three treatments in Trial-2^MP^ was not significantly different (F = 0.046; d.f. = 2; *p* < 0.956). The mean survival rate for each treatment in Trial-2^MP^ was A (*Amylostereum areolatum* mycelial suspension in SABAX pour water and autoclaved raw pine sawdust) (29.60 ± 6.111%), B (SABAX pour water and autoclaved raw pine sawdust) (27.00 ± 6.111%), and C (SABAX pour water) (28.45 ± 6.111%).

The data for Trial-2^WC^ (Figure 3A) was normally distributed according to the Shapiro–Wilk test (W = 0.941; *p* < 0.098). A one-way ANOVA showed that there was a significant difference between at least two of the groups in Trial-2^WC^ (F = 12.454; d.f. = 2; *p* < 0.001). The Tukey’s Honestly Significant Difference (HSD) post hoc test revealed that the mean survival rate in Trial-2^WC^ for Treatment A (44.20 ± 4.906%) was significantly different from Treatment B (70.20 ± 4.906%) (*p* = 0.002; 95% C.I. = [8.799, 43.20]) and Treatment C (77.00% ± 4.906%) (*p* = 0.002; 95% C.I. = [15.599, 50.00]). The mean survival rate did not differ significantly between Treatment B and Treatment C in Trial-2^WC^.

The pooled data from Trial-2^MP^ and Trial-2^WC^ (Figure 3B) were not normally distributed according to the Shapiro–Wilk test (W = 0.947; *p* < 0.012), which remained the case after data were transformed. Therefore, a Kruskal–Wallis test was performed, which showed that the mean overall survival rate between Trial-2^MP^ (28.35 ± 18.677%) and Trial-2^WC^ (63.80 ± 20.754%) was significantly different (ChiSquare = 27.117; d.f. = 1; *p* < 0.001).

### 3.3. Rearing of Sirex Noctilio Larvae, Emerging from Activated Eggs, on a Modified Artificial Diet

The quantities used in the modified artificial diet produced roughly 22 Petri dishes. A total of 286 and 970 larvae that emerged from activated eggs in Trial-2^MP^ and Trial-2^WC^, respectively, were subsequently reared on the modified artificial diet. In most cases, larvae from both trials survived past seven days; however, only a few survived past 28 days. For Trial-2^MP^, no larvae survived past 14 days. For Trial-2^WC^, 14 larvae survived past 40 days, six larvae past 50 days, three larvae past 61 days, two larvae past 86 days, and one larva until 92 days. Larvae were highly moisture sensitive and required an intermediate amount of water in the diet in order to survive. It was beneficial to keep larvae in the dark as it prevented excessive moisture loss. Larvae that died displayed various symptoms such as (i) excessive fluid filling the prothoracic segment (Figure 4G), (ii) darkening of the head capsule (Figure 4G,H), and (iii) appearance of a dark mass in the central cavity (Figure 4H). Specific observations were made for larvae from Trial-2^WC^. Small white globules, referred to as ‘fat bodies’, appeared under the integument and started to arrange in two slightly scattered parallel bands across the abdominal segments on the ventral and dorsal sides of larvae (Figure 4E). These fat bodies appeared in larvae around days 18 and 42 post-emergence. During this time, larvae would also start to obtain a yellowish colour in their cuticle, especially in the abdominal segments. As larvae developed, the size and number of fat bodies would increase, as well as the intensity of yellowish pigment in the cuticle (Figure 4C,D). Although factors such as age, female source, and diet, amongst others, were kept constant, size variation was observed in larvae at day 24 (Figure 4F).

## 4. Discussion

In this study, *Sirex noctilio* eggs have, for the first time, been successfully activated ex vivo using a novel technique requiring the application of mechanical pressure. The observed development of activated eggs corresponded with descriptions observed by Madden (1981) [33] and are the first photographic records available of *S. noctilio* egg development, from activation to emergence of first instar larvae. Our findings are further supported by research conducted in *Drosophila* and other hymenopteran insects, where eggs were activated using mechanical pressure [16,17,18,19,48]. In *Drosophila*, light pressure was applied to activate eggs by placing a coverslip on top of the eggs [48]. To our knowledge, this is the only study similar to the technique developed here. Mechanical pressure is thought to trigger egg activation due to the maternal nuclease being displaced, which prompts the cell to resume meiosis [14]. In addition, as a result of mechanical pressure, a rise in internal free calcium (Ca^2+^) levels (calcium wave) in activated eggs has been observed [3,48]. Mechanical pressure could cause an increase in Ca^2+^ since stretching or distorting the membrane of animal cells has been shown to result in increased intracellular Ca^2+^ levels [49,50]. Further investigation would be required to determine if mechanical pressure also causes a Ca^2+^ wave in *S. noctilio* eggs after activation.

This egg activation technique was repeatable for eggs from two climatically different regions in South Africa, indicating its promise for future work to obtain first instar S. *noctilio* larvae from female wasps via dissection. Successful activation of *S. noctilio* eggs is the first step in establishing a laboratory-rearing protocol, allowing continuous access to all life stages of *S. noctilio*. Genetic insect pest management strategies, for example, using Clustered Regularly Interspaced Short Palindromic Repeats/CRISPR associated 9 protein (CRISPR/Cas9), predominantly rely on microinjection to deliver the necessary components into eggs during a narrow time window before blastoderm formation [22]. By manually activating *S. noctilio* eggs, the timing of microinjection can be optimised to accurately target the pre-blastoderm phase. As microinjection damages insect eggs, resulting in a high mortality rate [51], large numbers of eggs are needed to increase the likelihood of producing a transformed individual. The developed technique grants access to a large number of eggs since a single *S. noctilio* female may contain up to 500 eggs [26], which can be activated simultaneously.

The present study has shown that eggs from the Western Cape had a significantly higher mean survival rate across all treatments (64%) compared to Mpumalanga (28%). We attribute the significant difference observed between the activated eggs from the two different regions to factors such as climatic region, female size, genetic diversity, or host species. The Lowveld of Mpumalanga (where the pine logs for Trial-2^MP^ were collected) generally has a higher mean temperature than the Western Cape [52]. Higher temperatures have been shown to cause a reduction in *S. noctilio* wasp size, and these females had normal, normal–small, and abnormal eggs [53]. Our observations agree with this study since adult females from Mpumalanga were generally smaller, and eggs appeared smaller compared to adult females and eggs from the Western Cape. Smaller eggs have reduced hatching success and are less likely to survive in some insects, including butterflies and honeybees [54,55]. This may contribute to the lower egg activation survival rate for eggs from Mpumalanga. A comparison of the genetic diversity within *S. noctilio* populations in South Africa has indicated that individuals from Mpumalanga are far less genetically diverse than individuals from the Western Cape [56]. Low genetic diversity can reduce the fitness of a population experiencing environmental stress since the capacity of the population to adapt to environmental change is limited [57]. Therefore, the lower genetic diversity in the Mpumalanga *S. noctilio* population may reduce the ability of eggs to withstand the immense mechanical pressure during egg activation and adapt to laboratory conditions. In southern Africa, *P. radiata* and *P. patula* are extensively planted in the winter rainfall and summer rainfall regions, respectively [58]. The *S. noctilio*-infested logs used in Trial-2^WC^ were *P. radiata* (24 years old), and for Trial-2^MP^ were *P. patula* (12–16 years old). Different *Pinus* species have been found to affect larval development and egg-hatching success for the pine caterpillar, *Dendrolimus punctatus*, (Lepidoptera: Lasiocampidae) [59] and may have contributed to the differences in egg activation survival rate observed in this study. It is uncertain whether host age would impact the life parameters of *S. noctilio*. These above-mentioned factors might influence the vigour of *S. noctilio* eggs from different populations in South Africa; however, further investigation is required to determine the extent to which these factors are important. Future research may explore the impact of these above-mentioned factors by, for example, conducting separate trials within a region, grouping females of the same size, and comparing the egg activation survival rate of females of the same size from the same host between regions.

Our results showed that *S. noctilio* eggs can develop and hatch in high moisture levels in the absence of *A. areolatum* in an artificial laboratory environment. In fact, the presence of *A. areolatum* had an adverse effect on egg survival in an artificial environment due to its overgrowth encapsulating eggs and hindering eclosion. *Amylostereum areolatum* provided no benefit for egg development in an artificial laboratory environment, and contamination by secondary microorganisms was more prevalent in treatments containing *A. areolatum*. The symbiotic fungus *A. areolatum* has been thought to play a crucial role in the egg and larval development of *S. noctilio* by creating a suitable environment inside the host tree [33,60]. Unfavourable conditions for *A. areolatum* growth in the host (tree) have been proposed to result in delayed *S. noctilio* egg hatching, which can be as long a year [33]. Sirex *noctilio* eggs only hatched when oviposition drills and adjacent xylem vessels were colonised by *A. areolatum* [34]. This phenomenon is thought to be due to the volatile by-products of *A. areolatum* necessary for initiating embryogenesis [33]. Additionally, *A. areolatum* decreases the wood moisture levels, which is more optimal for egg development [60]. The results from this study, therefore, challenge the assumption that *A. areolatum* is essential for *S. noctilio* development in an artificial laboratory environment.

*Sirex noctilio* larvae were reared for a maximum of 14 days and 92 days on an artificial diet for Trial-2^MP^ and Trial-2^WC^, respectively, in the absence of *A. areolatum*. The low survival rate for larvae from Trial-2^MP^ is likely due to the degradation of antimicrobial compounds in the diet, causing microbial growth that adversely affected larval survival since the prepared diet was stored at room temperature for 14 months before usage. In a similar diet for *Maruca vitrata* (Lepidoptera: Crambidae), antimicrobial compounds suppressed microbial growth for 8–10 days [61].

The symbiotic nature of the *Sirex-Amylostereum* interaction has led to the widely accepted idea that *A. areolatum* is essential for *S. noctilio* development [33,36,42]. However, little is known about the nutritional dynamics between *S. noctilio* larvae and *A. areolatum* [31]. The role *A. areolatum* plays in larval nutrition may also be dependent on its life stage [36]. First, and some second instar, larvae were found to feed solely on the fungus, while later larval instars fed on wood colonised by *A. areolatum* [34]. In this study, healthy larval development with the appearance of fat bodies was noted when larvae were reared on the artificial diet without *A. areolatum*. In insects, fat bodies are physiological tissue that stores nutrients in the form of glycogen, fat, and albuminoids [62]. Fat bodies are also involved in the synthesis of most circulating metabolites and hemolymph proteins [63]. As *S. noctilio* larvae mature, most of the body cavity becomes filled with fat bodies [64]. Therefore, we can conclude that the nutrients in the diet were adequate for development and in a form easily absorbed by *S. noctilio*.

A noticeable variation in size was observed between *S. noctilio* larvae obtained from the same female and reared under the same conditions. Some larvae varied greatly in size from as early as 24 days, even though all other conditions were the same (i.e., they were from the same female, same egg activation batch, Petri dish, and diet). In another study on *S. noctilio*, a size variation also occurred in larvae originating from the same site of oviposition drilling [34]. This can be ascribed to larvae being exposed to different moisture levels in the wood and, thus varying abundance of *A. areolatum* [34]. However, the genetic makeup of larvae and other environmental factors can also play a role in the size variation observed [42]. In our case, the moisture levels and external factors were exactly the same for larvae in the same Petri dish, and the size differences are thought to be largely due to either genetic or undefined/unknown factors.

Successful application of the newly developed egg activation technique and rearing of *S. noctilio* first instar larvae on the modified artificial diet require cognisance of a few recommendations. Firstly, the amount of water and number of eggs inside the double-slide compress is crucial. All eggs will burst if the water is too little or excessive and if a small group of eggs (10 or less) are activated. The correct amount of water depends on the number of eggs to be activated; i.e., 50 eggs between 30 and 50 µL was found to be optimal. It can be expected that approximately 5% of eggs will burst during activation. Secondly, cleaning organic material attached to eggs reduced the growth of fungi and bacteria during egg development. It is best to clean eggs before activation in order to minimise manual movement of eggs during development, as this may adversely affect egg survival. Thirdly, for optimal results, eggs should be obtained from live females that experience minimal stress and do not experience stress for a prolonged period of time. Fourthly, the rearing of eggs and larvae in complete darkness was beneficial due to a reduction in moisture loss. Complete darkness simulated the conditions eggs and larvae would experience naturally inside the host tree, but this would need to be confirmed with further work. Lastly, the shelf life of the diet is important to consider. We recommend keeping the diet at 4 °C for short-term storage (7 days maximum) and −20 °C for long-term storage (3 months maximum) and transferring larvae onto a fresh piece of diet every 7 days.

## 5. Conclusions

This is the first study to activate *S. noctilio* eggs dissected from a female wasp and the only study to have used a double-slide compress to activate any insect egg. The egg activation protocol developed here is a quick and easy method requiring no special or expensive equipment or skills. This egg activation protocol is efficient and allows for mass activation of eggs, whereas previous methods activated hymenopteran eggs individually. The egg activation protocol may be applicable to other hymenopterans or insects whose eggs have similar characteristics. The rearing protocol for *S. noctilio* larvae developed here can be useful to establish a laboratory-reared population of *S. noctilio*, which will allow opportunities to answer broad questions on biology and enable continuous research. The development of a genetic pest management strategy for *S. noctilio* is now a possibility as a result of the reported egg activation and rearing protocol.

## Figures and Tables

**Figure 1 insects-14-00931-f001:**
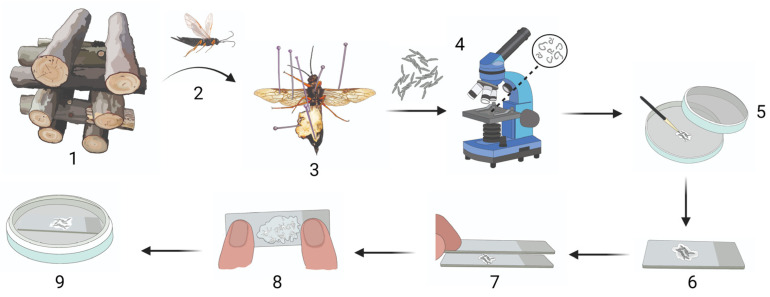
*Sirex noctilio* egg activation protocol. (**1**) Crossed-stacked *Sirex noctilio*-infested *Pinus* sp. logs stored in an outdoor double-netted cage from which (**2**) *S. noctilio* female wasps were collected after emergence and (**3**) dissected to remove eggs. (**4**) Eggs were checked for parasitism by nematodes, and unparasitised eggs were (**5**) cleaned on filter paper with a paintbrush. (**6**) Cleaned eggs were placed on a microscope slide in SABAX water, and (**7**) another slide was placed on top to create a double-slide compress. (**8**) A maximum downward force was applied to the double-slide compress for 1 min. (**9**) A single slide containing the activated eggs in SABAX water was placed in a Petri dish and wrapped in Parafilm. (Illustration created in BioRender.com. Illustration components created by Glenda Brits, Senior Graphic Designer, Department for Education Innovation, University of Pretoria, South Africa).

**Figure 2 insects-14-00931-f002:**
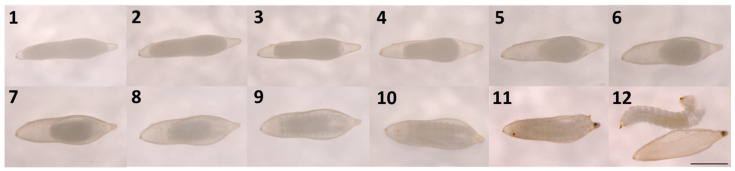
Development of a *Sirex noctilio* egg after activation from day 1 to day 12 (Scalebar: 1 mm).

**Figure 3 insects-14-00931-f003:**
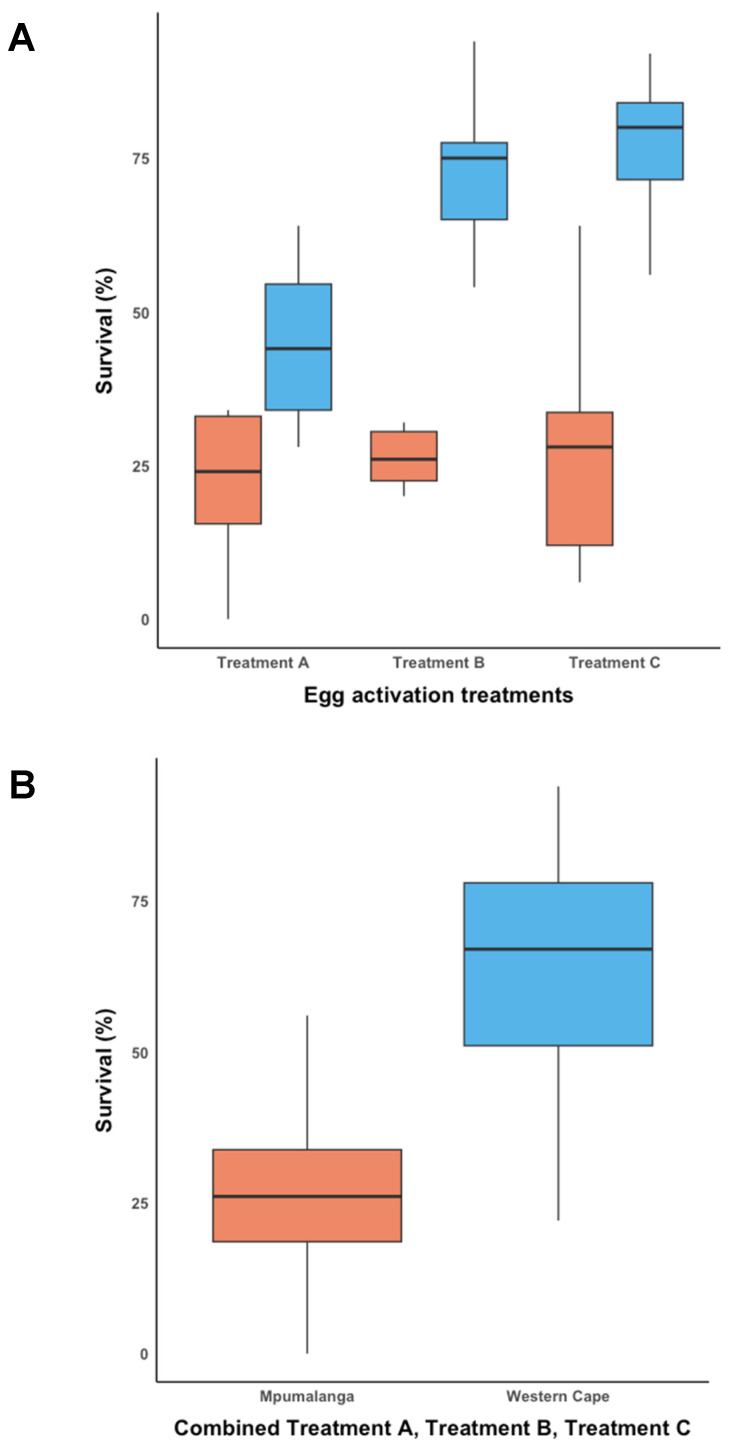
Egg activation survival rate for Mpumalanga (Trial-2^MP^) (indicated in orange) and Western Cape (Trial-2^WC^) (indicated in blue). (**A**) The egg activation survival rate across three treatments for Trial-2^MP^ and Trial-2^WC^, and (**B**) the overall egg activation survival rate for Trial-2^MP^ and Trial-2^WC^.

**Figure 4 insects-14-00931-f004:**
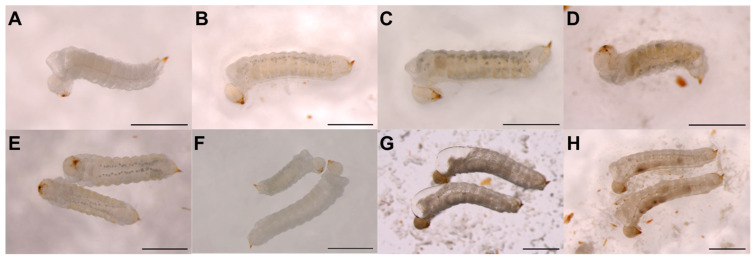
*Sirex noctilio* first instar larvae reared on an artificial diet. (**A**) newly hatched larva from activated egg; (**B**) 45 day old larva; (**C**) same larva as in B at 71 days old; (**D**) different larva as in (**B**) at 81 days old; (**E**) arrangement of fat bodies in two 33 day old larvae; (**F**) size difference between two 27 day old larvae from same female, activation batch and Petri dish with diet; (**G**) dead larvae with excessive fluid filling the prothoracic segment and a dark head capsule; (**H**) dead larvae with a dark mass in the central body cavity (Scalebar: 1 mm).

**Table 1 insects-14-00931-t001:** Ingredients for *Sirex noctilio* artificial diet.

Ingredient	Amount
Distilled water	257.81 mL
Agar	8.20 g
**Group 1**	
Raw wheat germ	19.92 g
Torula yeast powder	10.55 g
**Group 2**	
Wesson salt mixture	1.05 g
Sorbic acid	0.73 g
Methyl paraben	0.73 g
Sucrose	5.86 g
Casein from bovine milk	3.52 g
Sodium propionate	0.45 g
**Group 3**	
Cholesterol	0.35 g
Autoclaved wheatgerm oil	1.64 mL
**Group 4**	
Choline chloride	0.09 g
Vanderzant vitamin mixture for insects	1.55 g
Vitamin A beadlets	0.12 g
**Group 5**	
Alpha cellulose	45.70 g
**Group 6**	
Streptomycin sulfate salt	0.3 g
SABAX pour water	2 ml
**Group 7**	
Autoclaved pine sawdust	50 g

## Data Availability

The data presented in this study are available in Figshare at https://doi.org/10.6084/m9.figshare.24473938.v1, accessed on 2 November 2023.

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
