# Peer review of "Mechanical Egg Activation and Rearing of First Instar Larvae of Sirex noctilio (Hymenoptera: Siricidae)"

_insects, 2023, doi:10.3390/insects14120931_

Round 1

Reviewer 1 Report

Comments and Suggestions for Authors

Upon thorough review of the manuscript, the authors have presented a well-substantiated interpretation of their results, making a valuable addition to the field of entomology. I commend the authors for their meticulous approach and recommend the publication of this study. However, I have some suggestion. I hope these suggestion will strengthen the paper

Abstract: The abstract provides a clear overview of the study's objectives, methods, and findings. It sets the context effectively by mentioning previous studies on egg activation in Hymenoptera and introduces the subject of the study, Sirex noctilio, adequately. However, It might be beneficial to briefly mention the rationale behind choosing the two climatically different regions in South Africa for the study. This could provide readers with a better understanding of the study's design. The abstract could benefit from a sentence explaining why the study hypothesized the necessity of the symbiotic fungus for egg development. This would help readers grasp the significance of the finding that Amylostereum areolatum is not required for egg survival. Consider mentioning the broader implications of the finding that Amylostereum areolatum adversely affects egg eclosion. How might this impact the understanding of Sirex noctilio's biology or pest management strategies?

Introduction: The introduction provides a comprehensive background on egg activation, parthenogenesis, and the significance of these processes in Hymenoptera. It sets the stage well for the study by discussing the various methods of artificial egg activation in different insect species, which is relevant to the research being presented. While the introduction is comprehensive, it could benefit from a more concise presentation. Some details, such as the specific methods used in other hymenopteran species, could be summarized to maintain focus on the study's objectives. It may be helpful to briefly discuss the current understanding of the role of A. areolatum in the biology of S. noctilio to provide a clearer rationale for the hypothesis that the fungus is beneficial for egg development. Consider providing a brief overview of the potential genetic pest control methods mentioned (e.g., CRISPR/Cas9 gene drives, pgSIT) and how they relate to the need for a better understanding of S. noctilio's early life stages.

Material and method: The methods for insect collection and storage are detailed and well-described. The use of logs from two climatically different regions in South Africa is a good approach to ensure variability in the study. The storage conditions in the outdoor double-netted cage and the measures taken to maintain optimal conditions for larval and pupal development are adequately explained. However, It would be beneficial to include the rationale behind the selection of the specific regions in South Africa. This could provide additional context for the choice of study sites.

The method section could be enhanced by providing a brief justification for the choice of the specific mechanical pressure treatments. What previous studies or observations informed these choices?

Clarify the criteria used to determine the level of mechanical pressure applied. For instance, was a specific weight or force measurement used to ensure consistency in the application of "substantial mechanical pressure"?

Consider mentioning any steps taken to ensure the consistency of egg handling across different trials and treatments. This would strengthen the reliability of the results.

it would be beneficial to include a rationale for the specific amount of mechanical pressure applied. How was this determined to be the optimal amount for egg activation?

The preparation of the A. areolatum mycelial suspension could be described in more detail. For instance, how was the concentration of the suspension determined?

In Section 2.5, the modifications made to the A. glabripennis diet for S. noctilio could be further explained. What specific dietary needs of S. noctilio were considered in these modifications?

The method for transferring larvae onto the artificial diet could be clarified. How was it ensured that the larvae were not damaged during this process?

Statistical analysis: This provides a concise overview of the statistical analyses conducted for the trials. The choice of Shapiro-Wilk test for normality, ANOVA for comparing treatments, and Tukey’s HSD for post-hoc analysis is appropriate for the data described. It would be beneficial to include the sample sizes for each treatment group in Trial-2MP and Trial-2WC. This information is crucial for understanding the robustness of the statistical analyses.

The rationale for using a Kruskal-Wallis test for comparing overall survival rates between Trial-2MP and Trial-2WC should be elaborated. Specifically, why was this non-parametric test chosen over others?

The section could benefit from a brief explanation of why no statistical analyses were conducted for Trial-1. Was it solely due to the lack of survival in two treatments, or were there other considerations?

Results: The interpretation of the results in this study is scientifically robust, showcasing a commendable level of rigor and understanding.

Discussion: The discussion effectively highlights the novelty and significance of the study, particularly the successful ex vivo activation of Sirex noctilio eggs using mechanical pressure. The comparison with similar studies in Drosophila and other hymenopteran insects provides a good context for the findings. The discussion could benefit from a more detailed exploration of the potential mechanisms behind the observed effects of mechanical pressure on egg activation. While the displacement of maternal nuclease and the rise in Ca2+ levels are mentioned, a deeper dive into the literature could provide a more comprehensive understanding.

The section on the differences in egg survival rates between the Western Cape and Mpumalanga regions is informative. However, it would be strengthened by discussing potential experimental controls or future studies that could isolate and verify the impact of the factors mentioned (e.g., climatic region, female size, genetic diversity, host species).

The role of Amylostereum areolatum in the development of S. noctilio is a critical aspect of the discussion. While the study challenges the assumption of its necessity, it would be beneficial to discuss the implications of these findings for the understanding of Sirex-Amylostereum symbiosis and potential future research directions.

Reviewer 2 Report

Comments and Suggestions for Authors

This is a well written paper. I found no significant negative issues. I have made some editorial comments and changes using the comments tools, but overall I think the paper is in fine shape.

The biggest suggestions I have relate to the presentation of larval survival (in days) and larval size. Although the authors present differences in each of these in the body of the text, I think a figure would work well for the larval days of survival and a table would be nice for the variation in length/size of larvae (summarized in a manner of the authors choosing). However, I can't tell from the paper or the supplemental material if any measurements of the size of the larvae were taken, so a summary table may not be feasible if the measurements do not exist. However, I think adding a figure for the days larvae from different sources survive would be relatively easy. I urge the authors to consider adding these two elements if possible.

Round 2

Reviewer 1 Report

Comments and Suggestions for Authors

Authors modified paper according to comments. I am accepting this